# FEASIBILITY WITH LANGUAGE MODELS FOR OPEN-WORLD COMPOSITIONAL ZERO-SHOT LEARNING

## ABSTRACT

Humans can easily tell if an attribute (also called state) is realistic, i.e., feasible, for an object, e.g. fire can be *hot*, but it cannot be *wet*. In Open-World Compositional Zero-Shot Learning, when all possible state-object combinations are considered as unseen classes, zero-shot predictors tend to perform poorly. Our work focuses on using external auxiliary knowledge to determine the feasibility of state-object combinations. Our Feasibility with Language Model (FLM) is a simple and effective approach that leverages Large Language Models (LLMs) to better comprehend the semantic relationships between states and objects. FLM involves querying an LLM about the feasibility of a given pair and retrieving the output logit for the positive answer. To mitigate potential misguidance of the LLM given that many of the state-object compositions are rare or completely infeasible, we observe that significant work needs to go into exploiting the in-context learning ability of LLMs. We present an extensive study on many prompt variants and involving six LLMs, including two LLMs with open access to the logit values, identifying Vicuna and ChatGPT as best performing, and we demonstrate that our FLM consistently improves OW-CZSL performance across all three benchmarks.

## 1 INTRODUCTION

Humans have the ability to discern the feasibility of state-object pairs, effortlessly distinguishing between realistic and implausible combinations. For instance, while it is convincing for a *fire* to be *hot*, the notion of a *wet fire* is nonsensical. Open-world compositional zero-shot learning (OW-CZSL) (Mancini et al., 2021) seeks to emulate human-like understanding for compositional concepts. The task is to classify images to the correct state-object pair in the absence of explicit knowledge regarding the feasibility of the pairs in the candidate classes (referred to as open-world compositional zero-shot learning, OW-CZSL) when the model is trained with a small subset of feasible pairs. Models often struggle to achieve satisfactory performance in the open-world setting since all combinations of state-object pairs are considered prediction candidates, which includes both unseen feasible pairs and infeasible pairs, making the number of candidates much larger than the number of classes considered during training.

To address this challenge, prior works (Mancini et al., 2021; Karthik et al., 2022) proposed to remove possibly infeasible pairs from the label space using word vectors such as GloVe (Pennington et al., 2014) or using external resources such as ConceptNet (Speer et al., 2017). While these approaches represent a step forward, open-world compositional zero-shot learning remains extremely challenging as these approaches are limited in their capability to capture the semantic relationships underlying many rare concept compositions. Therefore, our goal is to propose a more effective approach for determining the feasibility of state-object pairs even if they are rare.

Large language models (LLMs) recently demonstrated strong language comprehension capabilities across various NLP tasks (Zhao et al., 2023). In this work, we propose Feasibility with Language Model (FLM) to predict the feasibility score of any state-object pair, with the purpose of better aligning with the human-annotated ground-truth feasibility than previous approaches. Concretely, we ask an LLM to give a binary response, i.e., "Yes" or "No", indicating the feasibility of the given state-object pair. The output logit for the positive answer would then be considered the feasibility score for the corresponding pair.

Inevitably, one challenge in using LLMs for feasibility prediction is that, provided without context a query could lead to many false negatives. Consider the "dark fire" class of the MIT-States (Isola et al., 2015) dataset, that is considered feasible. Asking a LLM, whether "dark fire" exists, yields the answer "No", presumably because the state "dark" is not typically associated with bright objects. However, "dark fire" is a reasonable class in MIT-States, as humans assigned this label to highlight the dark surroundings and dim visual theme for these images of fire. To teach the LLM about the relevant context for image classification, we can inform the LLM of semantically similar and feasible compositions from the training set, such as "dark lightning". As a result, the LLM can correctly infer *in-context* that the state "dark" can also be associated with "fire".

To summarize, our contributions are: 1) in Feasibility with Language Model (FLM) we propose to leverage LLMs to predict the feasibility of state-object pairs in open-world CZSL where we provide guided prompts that include examples of true feasible pairs, enhancing the LLMs' understanding of the CZSL task via in-context learning, 2) while the compositional label's feasibility judgement via FLM better align with human-annotated ground truth, FLM can also be integrated into any existing VLM, 3) FLM consistently improves CZSL performance over previous state-of-the-art methods on all three challenging benchmark datasets.

## 2 RELATED WORK

**CZSL.** CZSL aims to classify instances of state-object compositions not seen during training. Early approaches tackle CZSL by employing two separate classifiers for state and object primitives, followed by an additional module composing them to generate a classifier for the composition pairs (Misra et al., 2017; Purushwalkam et al., 2019). Another line of research treats the state as a transformation operator that modifies attributes of the object, and trains models to satisfy properties like symmetry or commutativity (Li et al., 2020; Nagarajan & Grauman, 2018). Additionally, GCNs have been leveraged to capture the relevance between states, objects, and state-object pairs, enhancing the models' generalization ability to unseen classes (Naeem et al., 2021; Ruis et al., 2021).

**VLMs for CZSL.** VLMs like CLIP (Radford et al., 2021) have been employed to address CZSL. CSP (Nayak et al., 2023) introduces a parameter-efficient learning technique, which fine-tunes the prompt, similar to CoOp (Zhou et al., 2022), but it updates the tokens representing states and objects instead of the prefixed context tokens. CSP utilizes vision-language understanding while adapting specifically to the downstream CZSL task, outperforming previous CNN-based task-specific methods.

**OW-CZSL.** Initially, CZSL models were evaluated exclusively on unseen classes. Then Chao et al. (2016) introduced a generalized setting that considers both seen and unseen classes as potential labels, which was first used in CZSL by Purushwalkam et al. (2019). The open-world setting (Mancini et al., 2021), extends the output space to include all possible combinations of states and objects, which is a more challenging task due to the substantial increase in the number of potential candidates. Consequently, identifying and disregarding infeasible state-object pairs becomes crucial. Prior approaches have employed word vectors, i.e., CompCos (Mancini et al., 2021), or external knowledge, i.e., KGSP (Karthik et al., 2022), to determine the feasibility of pairs. In contrast, we propose to leverage state-of-the-art large language models to capture the feasibility of pairs more effectively.

**LLMs as guidance.** LLMs have solved many NLP tasks, e.g. GPT-3 (Brown et al., 2020), Chat-GPT (Ouyang et al., 2022), PaLM 2 (Anil et al., 2023), Claude 2 (Anthropic, 2023) LLaMa 2 (Touvron et al., 2023a;b) and Vicuna (Chiang et al., 2023) demonstrate exceptional language understanding capabilities and have been widely applied across diverse NLP downstream tasks. It was shown by Brown et al. (2020) that LLMs can perform in-context learning without requiring fine-tuning, i.e., LLMs can significantly improve on a given task through task-specific exemplars demonstrating how a task is performed. On top of few-shot examples, providing explanations in-context further boost task performance on challenging tasks (Lampinen et al., 2022). Additional context can also be directly provided by the LLM. With chain-of-thought prompting (Wei et al., 2022), the in-context demonstrations provide more elaborate answers, resulting in the same behaviour of the LLM and ultimately more accurate responses. Similarly, Arora et al. (2023) find that aggregating multiple prompting structures helps alleviate the sensitivity of LLMs in-context prompt design.

**LLMs in vision tasks.** One vision application that uses LLMs is image classification through VLMs. For instance, Pratt et al. (2022) and Menon & Vondrick (2023) leverage LLMs to generate descriptions

for each class by querying the class name. Instead of relying solely on class names, these descriptions provide a better text representation, achieving improved performance in the image classification task. For visual question-answering (VQA) tasks, LLMs have been used as a knowledge base (Yang et al., 2022b) and Yang et al. (2022a) use class descriptions generated by GPT-3 as concept bottlenecks for interpretable image classification. In this work, we employ LLMs for the first time in OW-CZSL.

# 3 IN-CONTEXT FEASIBILITY PREDICTION FRAMEWORK

In this section, we first describe the general setting of compositional zero-shot learning in §3.1. Next, we explain the novel utilization of large language models for predicting feasibility scores in §3.2.

## 3.1 OPEN-WORLD COMPOSITIONAL ZERO-SHOT LEARNING (OW-CZSL)

CZSL aims to classify an image, where each class is a state-object combination. Given a set of states $\mathcal{S}$ and objects $\mathcal{O}$, the label space of a training set $\mathcal{Y}_{seen}$ corresponds to the subset of all possible pairs of state and object, $\mathcal{Y}_{seen} \subset \mathcal{Y}_{all}$, where $\mathcal{Y}_{all} = \{(s, o) | s \in \mathcal{S} \text{ and } o \in \mathcal{O}\}$. The model is trained on the training set with candidate labels as seen classes $\mathcal{Y}_{seen}$, and the goal of CZSL is to classify an image from the test label space $\mathcal{Y}_{test}$ that contains both seen and unseen classes, $\mathcal{Y}_{test} = \mathcal{Y}_{seen} \cup \mathcal{Y}_{unseen}$, where $\mathcal{Y}_{unseen}$ denotes the unseen classes during the training, $\mathcal{Y}_{seen} \cap \mathcal{Y}_{unseen} = \emptyset$ and $\mathcal{Y}_{unseen} \subset \mathcal{Y}_{all}$.

In the closed-world setting, it is assumed that we have prior knowledge of feasible sets, i.e., the label space at test time is restricted to $\mathcal{Y}_{test} \coloneqq \mathcal{Y}_{seen} \cup \mathcal{Y}_{unseen}$ and known to the model. The open-world setting assumes no prior information about the set of unseen compositions at test time, i.e., $\mathcal{Y}_{test} \coloneqq \mathcal{Y}_{all}$. The substantial increase in the label space in the open-world setting leads to a significant performance gap compared to the closed-world setting. To mitigate this performance gap, previous works (Mancini et al., 2021; Karthik et al., 2022) have developed a function $g(\cdot)$ that assigns a feasibility score to each class, indicating the likelihood of its feasibility. By setting a threshold $\tau$, classes with scores below the threshold are deemed infeasible and consequently removed from the test label space, $\mathcal{Y}_{test} \coloneqq \{y | y \in \mathcal{Y}_{all} \text{ and } g(y) \geq \tau\}$.

It is important to note that an accurate feasibility function plays a critical role in open-world CZSL. If the function assigns low feasibility scores to truly feasible classes, the model will fail to predict the correct class as it is absent from the test label space. Conversely, if the function assigns high feasibility scores to numerous infeasible classes, it becomes more likely that the model will make incorrect predictions.

## 3.2 FEASIBILITY WITH LANGUAGE MODEL (FLM)

LLMs are autoregressive models that generate words by sampling from a predicted probability distribution, i.e., they model $p_{\text{LLM}}(t_k | t_1, \ldots, t_{k-1})$ where $t$ is a token from the vocabulary of the language model. In other words, the output probability indicates how certain the LLM is that a given token should appear next. Essentially, our FLM uses the output of an LLM as a measure of feasibility.

**Canonical prompt.** To obtain feasibility scores using LLMs, we construct a prompt $c$ that consists of a system message, *sysmsg*, and a human message, *hmsg*. The *sysmsg* provides the LLM with general guidance while the *hmsg* asks the LLM to assess the feasibility of a given class.

$$c = \{ sysmsg : \text{``You are a helpful, respectful and honest assistant. Answer with a single word, yes or no.''},$$
$$hmsg : \text{``Does a/an \{s\} \{o\} exist in the real world?''} \}$$

where we refer to the first sentence of the *sysmsg* as the *persona* component (Salewski et al., 2023), the second as the *instruction* component, and the sentence of the *hmsg* as the *query* component. The placeholders $\{s\}$ and $\{o\}$ represent the state and object of the class, respectively.

The output probability distribution generated by the LLM reflects its level of certainty regarding the occurrence of specific words. In our case, the probability or the logit of the word "Yes" in the output distribution indicates the LLM's confidence in the feasibility of the given pair $(s, o)$. We interpret this output as a feasibility score. More formally, our feasibility score function is

$$g(s, o) = \log p_{\text{LLM}} (t = \texttt{"Yes"} | f(s, o; c)) \tag{1}$$

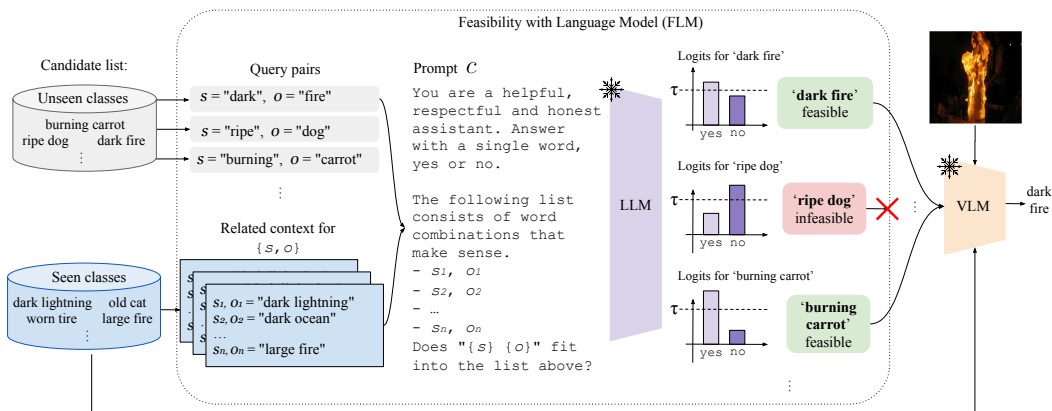

Figure 1: The pipeline of our Feasibility with Language Model (FLM) method. We constrict a prompt containing a list of related seen classes from the training set and a query to classify an unssen state-object pair as feasible. By comparing the LLM logit for the token "Yes" with a threshold $\tau$ we determine whether a pair is feasible, in which case it is used for OW-CZSL classification.

where $\log p_{\text{LLM}}$ indicates the unnormalized output logits and $f(s, o; c)$ denotes a function that composes the prompt $c$ with the target state-object pair $(s, o)$. To obtain a real-valued score, this method requires local access to the LLM. When an LLM is accessed through an API, such as ChatGPT, it might not expose the output probabilities or logits of the model, i.e. $\log p_{\text{LLM}}(t = \texttt{"Yes"}|f(s, o; c))$. In this case we can only retrieve a binary score of "Yes" or "No".

**In-context learning.** There is potential for incorrect responses when simply querying the LLM about the feasibility of a given pair, e.g. "dark fire" as mentioned in §1. Motivated by this, we leverage the in-context learning capabilities of LLMs. In-context learning enables LLMs to adapt to new tasks with minimal examples. In addition to the *query* component, we introduce a *guidance* component to the human message. The guidance includes a few examples of true feasible pairs, allowing the LLMs to learn from these instances and better understand what constitutes feasibility within the dataset. For example, a human message with guidance would be:

"`The following list consists of word combinations that make sense.`
  − $\{s_1\}\, \{o_1\}$
  − `...`
  − $\{s_n\}\, \{o_n\}$
 `Does "`$\{s\}\, \{o\}$`" fit into the list above?`"

where $\{(s_i, o_i)\}_{i=1}^n$ are few-shot examples of true feasible pairs. Several approaches can be employed to select examples for the guidance prompt. One straightforward method is to randomly sample pairs from the seen classes $\mathcal{Y}_{seen}$. Another approach is to leverage the information from the query pairs. Motivated by Mancini et al. (2021), we choose guidance pairs from the seen classes that either include the state $s$ or the object $o$, i.e., $\mathcal{Y}_{pos} = \{(s_i, o_i)|(s_i, o_i) \in \mathcal{Y}_{seen},\ s_i = s \text{ or } o_i = o\}$. This strategy enables the LLMs to gain a deeper understanding of the dataset-specific task within the in-context learning framework, improving predictions of the feasibility of query pairs. The overall pipeline is drawn in Figure 1. Our feasibility score function, denoted as Feasibility with Language Model (FLM), is formulated as

$$g(s, o) = \log p_{\text{LLM}}(t = \texttt{"Yes"}|f(s, o, \mathcal{Y}_{pos}; c)) \tag{2}$$

where $f(s, o, \mathcal{Y}_{pos}; c)$ denotes a function that composes the prompt $c$ with the target state-object pair $(s, o)$ and the related seen pairs $\mathcal{Y}_{pos}$.

**Versatility.** Once we obtain the feasibility scores for all combinations of pairs, the threshold $\tau$ determines the subset of all pairs that are deemed feasible. The infeasible pairs are discarded, and only the feasibile pairs are used as candidate labels for the VLM's prediction. Our feasibility scores can be integrated with any existing VLM to improve performance in the open-world setting.

## 4 EXPERIMENTS

We present our experimental findings on LLM-guided feasibility prediction in OW-CZSL. The experimental setup is detailed in §4.1, followed by a quantitative and qualitative comparison with baselines in §4.2. Furthermore, we evaluate the feasibility prediction in isolation in §4.3, compare a variety of LLMs in §4.4, and conduct an ablation study in §4.5, examining the different prompt components, such how the number of guidance examples influence the results.

### 4.1 EXPERIMENTAL SETUP

**Benchmarks.** We use three standard datasets for OW-CZSL, i.e., MIT-States (Isola et al., 2015), UT-Zappos (Yu & Grauman, 2014; 2017), and C-GQA (Naeem et al., 2021). Each dataset comprises a set of states and objects, where an object-state combination forms a class. MIT-States consists of 115 states and 245 objects, resulting in a total of 28,175 possible pairs. Among these possible pairs, 1,262 and 700 pairs are seen classes and unseen classes, respectively. UT-Zappos includes 16 states and 12 objects, leading to 192 possible pairs, with 83 seen classes and 33 unseen classes. Finally, C-GQA has 413 states and 674 objects, resulting in 278,362 possible pairs, with 5,592 seen classes and 1,963 unseen classes.

**Evaluation metric.** We follow the protocol of Purushwalkam et al. (2019) for OW-CZSL. Since the VLM is trained only on seen classes, it is prone to being biased towards classifying an image as one of the seen classes at test time. Concretely, a calibration bias is subtracted from the model outputs of the seen classes, and then the class is predicted. The calibration bias is varied to get the best combination of seen class accuracy (denoted as S), unseen class accuracy (U), harmonic mean of accuracy on seen class and unseen class (H), and area under the curve of seen class and unseen class accuracy (AUC). By tackling the feasibility prediction of unseen classes, we focus on improving the more challenging metrics (U, H, AUC) while the seen class accuracy (S) remains unaffected.

**Implementation details.** For OW-CZSL, hyperparameters are traditionally explored, and the best model is chosen based on the highest unseen validation accuracy. We perform a grid search on sentence variations as well as choose the threshold $\tau$ that determines whether the query class is feasible by the best unseen validation accuracy (more details in the Appendix). For the LLM, we use the Vicuna-13B model (Chiang et al., 2023) for the experiments unless otherwise indicated.

**Feasibility baselines.** We compare with GloVe embeddings (Pennington et al., 2014) as used in CompCos (Mancini et al., 2021) and CSP (Nayak et al., 2023), and the ConceptNet (Speer et al., 2017) as used in KGSP (Karthik et al., 2022). For GloVe, the cosine similarity between the concepts of the same primitives are calculated and merged to represent the feasibility score. ConceptNet (Speer et al., 2017) is a knowledge graph connecting words to obtain the feasibility scores which are calculated by the cosine similarities of ConceptNet embeddings.

### 4.2 LLM-GENERATED FEASIBILITY IN OW-CZSL

We evaluate our FLM method using three VLMs: CLIP (Radford et al., 2021), CoOp (Zhou et al., 2022), and CSP (Nayak et al., 2023), which are CLIP-based models. We choose VLMs since they outperform the CNN-based task-specific CZSL methods, as reported by Nayak et al. (2023). We use ViT-L/14 as the VLMs' backbone and run the CSP official code[1] with fixed hyperparameter settings for CoOp and CSP (details in Appendix), although optimizing these hyperparameters as done by Nayak et al. (2023) could further yield improvements. We run CoOp and CSP with 5 different seeds and report the mean and the standard deviation. CLIP is applied without any fine-tuning, and thus standard deviation is not reported.

**Quantitative comparison.** The experimental results are presented in Table 1 where we improve OW-CZSL performance across various scenarios and on all datasets. We first observe that on the MIT-States dataset, our FLM method achieves the highest harmonic mean of 17.4% and AUC of 5.76%, surpassing the GloVe feasibility function which shows 16.4% and 5.12%, and the ConceptNet feasibility function with 15.5% and 4.74% using the CSP model. FLM exhibits even more significant improvements on the UT-Zappos dataset. When compared to the GloVe feasibility, we observe a substantial 7.4% increase in AUC (from 22.6% to 30.0%) for the CSP model, a 6.3% increase

---

[1] https://github.com/BatsResearch/csp

| ViT-L/14 | | MIT-States | | | | UT-Zappos | | | | C-GQA | | | |
|---|---|---|---|---|---|---|---|---|---|---|---|---|---|
| VLM | Method | S | U | H | AUC | S | U | H | AUC | S | U | H | AUC |
| | GloVe | 30.21 | 14.6 | 13.0 | 3.10 | 10.8 | 19.3 | 10.6 | 1.60 | 7.59 | **3.92** | 2.46 | **0.20** |
| CLIP | ConceptNet | —"— | 12.5 | 12.7 | 2.75 | —"— | 21.3 | 10.3 | 1.69 | —"— | 2.01 | 2.59 | 0.13 |
| | FLM (ours) | —"— | **16.1** | **13.7** | **3.38** | —"— | **23.6** | **11.5** | **1.94** | —"— | 2.62 | **2.82** | 0.16 |
| | GloVe | 38.2±0.9 | 16.7±0.3 | 16.2±0.4 | 4.78±0.2 | 61.2±1.8 | 36.7±2.4 | 34.2±3.7 | 18.1±2.8 | 26.9±1.6 | 5.09±0.5 | 6.16±0.5 | 1.03±0.1 |
| CoOp | ConceptNet | —"— | 14.5±0.3 | 15.6±0.3 | 4.36±0.1 | —"— | 42.1±2.3 | 36.6±3.2 | 20.6±2.5 | —"— | 4.14±0.5 | 5.88±0.7 | 0.92±0.2 |
| | FLM (ours) | —"— | **18.7±0.3** | **17.4±0.5** | **5.40±0.1** | —"— | **49.6±1.7** | **40.6±3.1** | **24.4±2.7** | —"— | **5.16±0.3** | **6.91±0.4** | **1.13±0.1** |
| | GloVe | 45.1±0.9 | 14.9±0.3 | 16.4±0.4 | 5.12±0.2 | 62.8±0.9 | 45.8±1.8 | 38.9±0.8 | 22.6±1.0 | 30.2±0.5 | **4.58±0.5** | 6.12±0.5 | 1.09±0.1 |
| CSP | ConceptNet | —"— | 13.4±0.8 | 15.5±0.5 | 4.74±0.3 | —"— | 54.0±1.7 | 43.3±0.9 | 26.9±1.1 | —"— | 1.31±0.1 | 2.25±0.3 | 0.34±0.0 |
| | FLM (ours) | —"— | **16.6±0.3** | **17.4±0.6** | **5.76±0.2** | —"— | **56.7±1.3** | **43.9±0.9** | **30.0±1.1** | —"— | 4.55±0.5 | **6.55±0.5** | **1.13±0.1** |

Table 1: CSZL results comparing Glove, ConceptNet and our FLM (Vicuna, logit) on MIT-States, UT-Zappos and C-GQA. We report seen (S) and unseen class accuracy (U), harmonic mean (H) and AUC using the CLIP, CoOp and CSP as base models. Ditto (—"—) denotes "same as above".

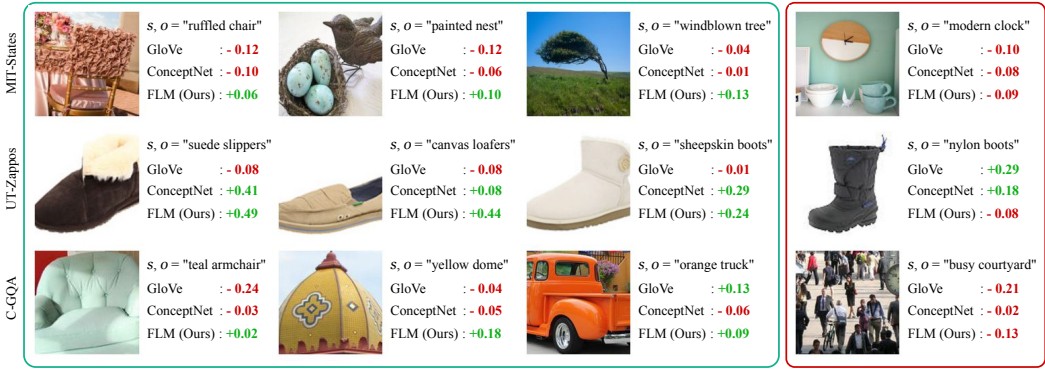

Figure 2: Feasible examples from the unseen test set along with feasibility scores normalized such that the threshold $\tau$ is at 0. Positive scores (green) indicate a correct prediction as feasible, while negative scores (red) incorrectly infer infeasibility of a pair. Red box includes failure cases of FLM.

(from 18.1% to 24.4%) with the CoOp model, and a 0.34% increase (from 1.60% to 1.94%) in the CLIP model. Similarly, FLM outperforms ConceptNet on UT-Zappos. On the C-GQA dataset, FLM performs the best everywhere except on U and AUC metrics in the CLIP model. However, CLIP is not the best model for OW-CZSL and generally falls behind CoOp and CSP. Overall, FLM achieves the best results on all metrics across all datasets. These results indicate that FLM can better differentiate between feasible state-object pairs and infeasible ones, as it facilitates all base OW-CZSL models to obtain a higher score, closing the gap to the closed-world setting.

**Qualitative comparison.** In Figure 2, we show qualitative results of feasible images from the unseen classes alongside the absolute difference of the feasibility score from the threshold. Positive values (green) indicate a correctly identified pair, while negative values (red) indicate an incorrect feasibility prediction. For each dataset, we show examples comparing FLM with GloVe and ConceptNet. For instance, FLM correctly identifies that "ruffled chair" is feasible for the MIT-States dataset, and that "teal armchair" is feasible in the context of the C-GQA dataset, both of which are considered infeasible by GloVe and ConceptNet. By providing seen pairs that are relevant to the query pair in the guidance prompt, e.g. "ruffled bed" for the query "ruffled chair" and "tan armchair" for the query "teal armchair", our FLM correctly identifies the given query pairs as feasible.

## 4.3 Ablation Study: Feasibility Prediction in Isolation from the OW-CZSL Task

We evaluate the feasiblity prediction in isolation from the OW-CZSL task and analyze the distributions of feasibility scores on the C-GQA dataset. In Figure 3, the unseen classes are referred to as "feasible classes" (blue) and the classes that are absent in both the seen and unseen sets are referred to as "confusing classes" (orange). Note that the scores obtained from each method have been normalized to fall within the range of 0 and 1.

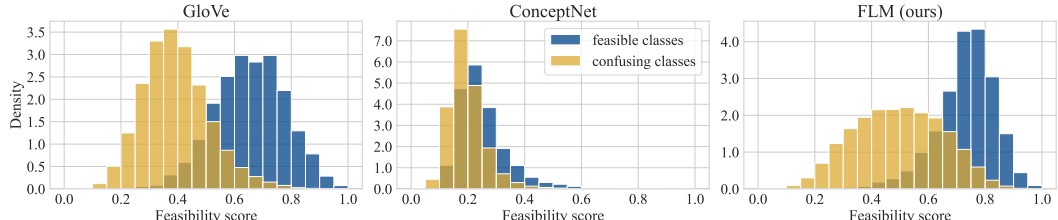

Figure 3: Distributions of feasibility scores of all state-object pairs. For best separation, feasible classes should be close to 1 and all remaining confusing classes close to 0.

|  | MIT-States | | | | UT-Zappos | | | | C-GQA | | | |
|---|---|---|---|---|---|---|---|---|---|---|---|---|
|  | Feas. acc. | Infeas. acc. | Arith. mean | H. mean | Feas. acc. | Infeas. acc. | Arith. mean | H. mean | Feas. acc. | Infeas. acc. | Arith. mean | H. mean |
| GloVe | 51.7 | **93.2** | 72.5 | 66.5 | 78.8 | 38.2 | 58.5 | 51.4 | 40.0 | **98.5** | 69.2 | 56.9 |
| ConceptNet | 52.0 | 92.5 | 72.3 | 66.6 | **100.0** | 13.2 | 56.6 | 23.3 | 26.3 | 91.7 | 59.0 | 40.9 |
| FLM (ours) | **64.7** | 86.1 | **75.4** | **73.9** | 93.9 | **46.1** | **70.0** | **61.8** | **70.9** | 89.2 | **80.1** | **79.0** |

Table 2: Accuracy of correctly identifying feasible and infeasible open-world pairs from $\mathcal{Y}_{all}$ using the same threshold $\tau$ as in Table 1. FLM uses Vicuna (logit) as LLM.

FLM exhibits a better separation between the two distributions than GloVe and ConceptNet implying that our approach more effectively distinguishes between feasible and infeasible classes, providing an accurate assessment of the feasibility. It is important to note that many combinations contained in the "confusing classes" can still be realistically feasible, but simply not included in the dataset. Therefore, is is unlikely these distributions can be perfectly separated. By employing an appropriate threshold, our FLM method includes a greater number of feasible classes while significantly reducing the inclusion of infeasible classes among the candidate labels for open-world CZSL.

We evaluate overlap of the feasibility scores with the human annotations quantitatively in Table 2. For every state-object pair in $\mathcal{Y}_{all}$, we compute the feasibility prediction of GloVe, ConceptNet, and FLM and compare it to the human-annotated ground truth of feasible classes $\mathcal{Y}_{unseen}$. We report feasibility accuracy as the ratio of pairs in $\mathcal{Y}_{unseen}$ correctly identified as feasible and, analogously, infeasible accuracy for the ratio of all other classes predicted as infeasible. As we have seen above, the distributions of feasible and infeasible pairs are not perfectly separable and, thus, these two metrics form a trade-off that can be varied using the threshold value $\tau$. The reported metrics are calculated using the same threshold values as in Table 1. We observe that our FLM performs the best on either feasible accuracy or infeasible accuracy, which suggests that the best trade-off between feasible and infeasible accuracy varies by dataset. Considering both metrics together through arithmetic and harmonic means, our method performs the best across all datasets, often by a significant margin, such as improving harmonic mean over GloVe by +10.4% on UT-Zappos and +22.1% on C-GQA. Consequently, the more accurate prediction of feasibility scores, which better align with the human-annotated ground truth, results in FLM performing better than GloVe and ConceptNet.

## 4.4 ABLATION STUDY: COMPARING LARGE LANGUAGE MODELS

We use ChatGPT (Ouyang et al., 2022; Brown et al., 2020), GPT-4 OpenAI (2023), PaLM-2 Anil et al. (2023), Claude-2 (Anthropic, 2023), LLaMa-2-Chat-13B (Touvron et al., 2023b) and Vicuna-13B (Chiang et al., 2023). Vicuna-13B and LLaMa-2-Chat-13B are open-source language models fine-tuned to follow instructions that reach similar capabilities to ChatGPT in some settings. For brevity, we refer to these models as "Vicuna" and "LLaMa-2" in the following. One advantage of Vicuna and LLaMa-2 over proprietary models is the accessibility of internal values such as the probability or logit of the output words. Specifically, we utilize the logit value of the word "Yes" as our feasibility score. Moreover, we compare with ChatGPT, GPT-4, PaLM-2 and Claude-2 as proprietary LLMs where we query the API to obtain binary feasibility scores, i.e., a score of 1 when the model answers with "Yes" and 0 when it answers with "No". In this case, the threshold $\tau$ cannot be varied and is set to 0.5.

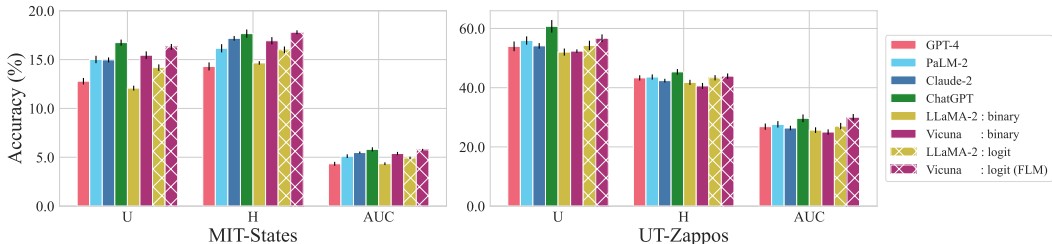

Figure 4: Comparison of FLM using Vicuna, LLaMA-2, and proprietary models (GPT-4, PaLM-2, Claude-2, and ChatGPT) as LLMs on MIT-States and UT-Zappos. Proprietary models can only provide a binary "Yes" or "No" response, whereas for Vicuna and LLaMA-2 we evaluate both the binary and logit outputs as feasibility scores.

| | | MIT-States | | | UT-Zappos | | | C-GQA | | |
|---|---|---|---|---|---|---|---|---|---|---|
| Prompt | | U | H | AUC | U | H | AUC | U | H | AUC |
| Canonical | | $13.5_{\pm0.4}$ | $15.4_{\pm0.3}$ | $4.70_{\pm0.2}$ | $47.1_{\pm1.4}$ | $39.2_{\pm0.6}$ | $23.1_{\pm0.6}$ | $2.16_{\pm0.5}$ | $3.40_{\pm0.6}$ | $0.52_{\pm0.1}$ |
| Instruction: hmsg begin | | $15.5_{\pm0.4}$ | $16.6_{\pm0.3}$ | $5.32_{\pm0.2}$ | $58.2_{\pm1.2}$ | $44.1_{\pm0.6}$ | $28.3_{\pm0.7}$ | $3.06_{\pm0.4}$ | $4.42_{\pm0.3}$ | $0.69_{\pm0.0}$ |
| Instruction: hmsg last | | $14.8_{\pm0.3}$ | $16.4_{\pm0.3}$ | $5.11_{\pm0.1}$ | $56.6_{\pm1.2}$ | $43.8_{\pm0.9}$ | $27.9_{\pm1.1}$ | $2.84_{\pm0.1}$ | $4.23_{\pm0.3}$ | $0.66_{\pm0.1}$ |
| Format | QA: yes | $10.3_{\pm0.7}$ | $12.0_{\pm0.5}$ | $3.39_{\pm0.3}$ | $53.1_{\pm1.6}$ | $42.4_{\pm1.3}$ | $26.7_{\pm1.2}$ | $2.23_{\pm0.3}$ | $3.40_{\pm0.6}$ | $0.49_{\pm0.1}$ |
| | QA: score | $10.8_{\pm0.3}$ | $12.3_{\pm0.3}$ | $3.53_{\pm0.1}$ | $54.2_{\pm1.6}$ | $43.4_{\pm0.9}$ | $26.9_{\pm1.0}$ | $2.36_{\pm0.1}$ | $3.69_{\pm0.1}$ | $0.54_{\pm0.0}$ |
| In-context | $\mathcal{Y}_{pos}$, 5 | $13.8_{\pm0.3}$ | $15.3_{\pm0.3}$ | $4.74_{\pm0.1}$ | $48.2_{\pm8.0}$ | $39.9_{\pm4.0}$ | $24.0_{\pm4.2}$ | $3.16_{\pm0.3}$ | $4.67_{\pm0.6}$ | $0.76_{\pm0.1}$ |
| | $\mathcal{Y}_{pos}$, 20 | $14.7_{\pm0.6}$ | $16.3_{\pm0.3}$ | $5.12_{\pm0.1}$ | $56.7_{\pm1.3}$ | $43.9_{\pm0.9}$ | $30.0_{\pm1.1}$ | $2.66_{\pm0.5}$ | $4.00_{\pm0.7}$ | $0.63_{\pm0.1}$ |
| | $\mathcal{Y}_{pos}$, 50 | $16.6_{\pm0.3}$ | $17.4_{\pm0.6}$ | $5.76_{\pm0.1}$ | —"— | —"— | —"— | $3.25_{\pm0.5}$ | $4.62_{\pm0.7}$ | $0.76_{\pm0.1}$ |
| | $\mathcal{Y}_{pos}$, 200 | —"— | —"— | —"— | —"— | —"— | —"— | $4.55_{\pm0.5}$ | $6.55_{\pm0.5}$ | $1.13_{\pm0.1}$ |
| | random, 200 | $13.2_{\pm0.6}$ | $14.9_{\pm0.7}$ | $4.50_{\pm0.3}$ | $50.4_{\pm0.9}$ | $40.7_{\pm0.7}$ | $24.8_{\pm0.7}$ | $2.95_{\pm0.3}$ | $4.51_{\pm0.6}$ | $0.72_{\pm0.1}$ |
| FLM (ours) | | $\mathbf{16.6}_{\pm0.3}$ | $\mathbf{17.4}_{\pm0.6}$ | $\mathbf{5.76}_{\pm0.1}$ | $\mathbf{56.7}_{\pm1.3}$ | $\mathbf{43.9}_{\pm0.9}$ | $\mathbf{30.0}_{\pm1.1}$ | $\mathbf{4.55}_{\pm0.5}$ | $\mathbf{6.55}_{\pm0.5}$ | $\mathbf{1.13}_{\pm0.1}$ |

Table 3: Ablation study. Experiments are done with CSP as VLM model. We ablate the canonical query without in-context exemplars, placing instruction component in the human message, a different format of the guidance component, varying the number of in-context pairs, and a random ordering of in-context examples. Ditto (—"—) denotes the result is the same as the previous line.

Proprietary LLMs such as ChatGPT (Ouyang et al., 2022; Brown et al., 2020) often demonstrate superior performance compared to open-source models such as Vicuna (Chiang et al., 2023) and LLaMa-2. However, there are distinctions between the two types of models in terms of accessibility. To ensure consistency and eliminate randomness, we set the temperature parameter to 0 during these experiments. Due to API constraints, we conduct these experiments only on the MIT-States and UT-Zappos datasets. The results for GPT-4, PaLM-2, Claude-2, ChatGPT, LLaMa-2, and Vicuna are depicted in Figure 4.

Across both datasets, we observe consistent trends. Firstly, both Vicuna and LLaMa-2 show lower performance with a binary answer compared to using logits on all three evaluation metrics. This suggests that accessing the logits provides valuable information for estimating feasibility. Among these two models, Vicuna outperforms LLaMa-2 clearly. Secondly, among the proprietary LLMs, ChatGPT performs best, surpassing PaLM-2, Claude-2 and even GPT-4. The differences are more pronounced on MIT-states than on UT-Zappos where most LLMs tend to perform similarly. Lastly, ChatGPT with a binary answer consistently outperforms Vicuna with a binary answer and oftentimes achieves even better results than Vicuna using logits. From these findings, we speculate that ChatGPT with logit access would likely surpass Vicuna with logit considerably, implying that more advanced LLMs with logit access would yield improved feasibility scores and, thus, could further push state-of-the-art in OW-CZSL.

## 4.5 ABLATION STUDY: ANALYSIS OF LLM-PROMPTS

**Comparison of instruction prompts.** We investigate the impact of the prompt by comparing the performance of our in-context learning prompt with two ablations: 1) the canonical prompt described

in §3.2 which does not use in-context examples, and 2) placing instruction component, e.g. "`Answer with a single word, yes or no.`", in the human message instead of the system message. The results on the three datasets are presented in the first three rows of Table 3.

Across all datasets, we observe that using the canonical prompt significantly drops the performance, e.g. 17.4%→15.4%, 43.9%→39.2%, 6.55%→3.40% in harmonic mean on MIT-States, UT-Zappos, and C-GQA, respectively. This highlights the importance of providing in-context guidance in our FLM for feasibility prediction. Moreover, placing an instruction in the human message, whether at the beginning or end, drops the performance on MIT-States and C-GQA while showing similar performance on UT-Zappos. This suggests that incorporating an instruction in the system message effectively guides the LLMs to the desired behavior.

**Format for in-context learning.** Arora et al. (2023) have demonstrated the effectiveness of LLMs' in-context learning when using a question-and-answer format. To investigate this approach for FLM, we use a question-answer format for the guidance prompt instead of a list. Specifically, we employ the guidance prompt "`Does a/an` $\{s_i\}$ $\{o_i\}$ `exist in the real world?  Yes.`", which is repeated for every related seen pair in $\mathcal{Y}_{pos}$, and the query prompt "`Does a/an` $\{s\}$ $\{o\}$ `exist in the real world?`" while keeping the rest of the process the same. The results are shown in the "Format QA: yes" row of Table 3.

We observe the performance drops across datasets, e.g. 17.4%→12.0 and 6.55%→3.69% in harmonic mean on MIT-States and C-GQA. The lower performance originates from this prompt format biasing the LLM to answer "Yes" because we only have access to feasible pairs. We observe a similar trend for "Format QA: score" where we use the same question-answer format, but instruct the LLM to respond with an integer score indicating the level of feasibility (see Appendix for details). Both of these results indicate that employing our proposed list format is crucial in obtaining accurate feasibility scores because we do not have access to infeasible examples.

**Number of pairs for in-context guidance.** To analyze the influence of the number of in-context examples in the guidance prompt, we conducted experiments varying the number of positive pairs. Recall that our FLM method selects related pairs in the guidance as $\mathcal{Y}_{pos} = \{(s_i, o_i) | (s_i, o_i) \in \mathcal{Y}_{seen}, \ s_i = s \text{ or } o_i = o\}$. The performance results are presented in the "in-context" rows of Table 3.

Across all datasets and evaluation metrics, performance consistently improves as the number of pairs in the guidance increases. For instance, on MIT-States, the harmonic mean increases from 15.3% to 16.3%, and subsequently to 17.8%, as the number of pairs expands from 5, to 20, and to 50, respectively. Each dataset contains a different maximum number of related seen pairs. Thus, performance does not improve beyond 50 for MIT-States and beyond 20 for UT-Zappos. Moreover, using up to 200 randomly selected state-object pairs results in worse performance than just 5 related pairs from $\mathcal{Y}_{pos}$ on MIT-States and C-GQA. This suggests that it is important to provide relevant in-context pairs and that more few-shot examples allow the LLM to better comprehend the context-dependent task, leading to more accurate feasibility scores.

## 5   CONCLUSION

In this paper, we proposed a novel approach that leverages large language models (LLMs) to predict the feasibility of the state-object pair for the open-world compositional zero-shot learning (OW-CZSL). By leveraging the autoregressive nature of LLMs, we designed prompts to query the feasibility of class pairs to LLMs, and obtain the output of the word "Yes" which we consider as feasibility score. We used the in-context learning capabilities of LLMs by providing guidance prompts that included a few examples of true feasible pairs. Our experimental results validated the effectiveness of our LLM-guided feasibility approach. We compared our FLM method with previous approaches and achieved better performance in MIT-States, UT-Zappos, and C-GQA datasets. Through an analysis of the feasibility score prediction, we demonstrated that our Feasibility with Language Model effectively differentiated between feasible and infeasible classes when compared to human-annotated ground truth. Furthermore, ablation studies on the prompt setting revealed that the in-context learning framework with seen pairs as a guide was a key factor to having high-quality feasibility scores.

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

APPENDIX

## A    BROADER IMPACT AND LIMITATIONS

Determining the feasibility of state-object pairs in all combinations is crucial when deploying the model. By accurately assessing feasibility, we prevent the model from predicting unrealistic classes, thus improving the model performance and reducing negative impacts on end-users. However, our FLM is limited in that we use prior knowledge, i.e. seen classes from the training dataset, which could be biased. If the seen classes represent only part of the semantics, e.g. only the texture-related attributes like "furry" or "zigzag" for animal in the seen classes while age-related states exist in the test set, our method may struggle to predict the true feasible pairs accurately. The bias can also lead to fairness issues if the seen classes only represent certain groups. Therefore, it is important to curate an unbiased seen class set to support the model's understanding of feasibility for the query pairs.

## B    PROMPT SEARCH IN FLM

There are four prompt components in FLM: persona, instruction, guidance, and query. In our experiments, We keep the *persona* component fixed as "`You are a helpful, respectful and honest assistant.`". We conduct a grid search using four instruction, four guidance, and four query sentences, which are:

$instruction\_list = \{$

  "`Answer with a single word, yes or no.`",

  "`Answer with a single word, yes or no, followed by an explanation.`",

  "`Answer with yes or no.`",

  "`Answer with yes or no, followed by an explanation.`",

$\},$

$guidance\_list = \{$

  "`The following list consists of words that fit together.`",

  "`The following list consists of word combinations that make sense.`",

  "`The given list consists of word combinations that make sense.`",

  "`The given list comprises word combinations that make sense.`",

$\},$

$query\_list = \{$

  "`Considering the list above, does "`$\{s\}\{o\}$`" fit into the list?`",

  "`Does "`$\{s\}\{o\}$`" fit into the list above?`",

  "`Does "`$\{s\}\{o\}$`" align with the contents of the list provided above?`",

  "`Considering the list above, does "`$\{s\}\{o\}$`" align with the contents?`",

$\},$

and select the combination that yields the highest unseen validation accuracy following the validation protocol and split of Nayak et al. (2023). On MIT-States, we use "`Answer with a single word, yes or no, followed by an explanation.`" as instruction, "`The following list consists of words that fit together.`" as guidance, and "`Does "`$\{s\}\{o\}$`" fit into the list above?`" as query. On the UT-Zappos, we use "`Answer with a single word, yes or no.`" as instruction, "`The given list consists of word combinations that make sense.`" as guidance, and "`Considering the list above, does "`$\{s\}\{o\}$`" fit into the list?`" as query. Finally on the C-GQA, we use "`Answer with a single word, yes or no, followed by an explanation.`" as instruction, "`The given list consists of word combinations that make sense.`" for CLIP and "`The given list comprises word combinations that make sense.`" for CoOp and CSP as guidance, and "`Does "`$\{s\}\{o\}$`" align with the contents of the list provided above?`" as query.

To examine a broader range of prompt variations, we report the results on MIT-States as a box plot in Figure 5. We first observe the results vary with different prompts. For instance, the Harmonic mean accuracy in CSP ranges from 15.5% to 18.1%. However, despite this variability, most of the prompts outperform the baselines. For instance, the Glove result always lies close to or below the lower quartile (25th percentile) of the box plot, and ConceptNet even further below that.

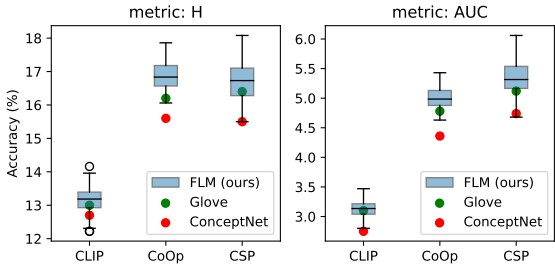

Figure 5: Prompt variation results on MIT-States dataset.

## C    HYPERPARAMETER SETTINGS IN CZSL MODELS

We train CoOp (Zhou et al., 2022) and CSP (Nayak et al., 2023) with the CSP official code [2] to get the fine-tuned models. Following the original CSP setting, we fine-tune a pretrained CLIP (Radford et al., 2021) model with a ViT-L/14 backbone for 20 epochs and choose the checkpoint with the highest validation accuracy on unseen classes. During training, we employ a batch size of 64, a learning rate of 5e-4, and a weight decay of 1e-5. Additionally, we set the attribute dropout rate for CSP to 0.3. We did not optimize these hyperparameters as originally done by Nayak et al. (2023). The model training is performed on a single A100 GPU. Similarly, when querying Vicuna-13B for our FLM, we use a single A100 GPU.

## D    QUESTION-ANSWER FORMAT: 0-9 SCORE

As mentioned in the main text, we observe that the prompt ablation "Format QA: yes" drops performance across all datasets, e.g. 17.4%→12.0 and 6.55%→3.69% in harmonic mean on MIT-States and C-GQA. The lower performance originates from the tendency of the LLM to answer "Yes" since the provided examples in the guidance are always positive. This phenomenon is also evident in Figure 6, where both the distributions of feasible and confusing classes overlap and lean closer towards 1. These observations indicate that employing a list format for guidance rather than a question-answer format is crucial in obtaining accurate feasibility scores.

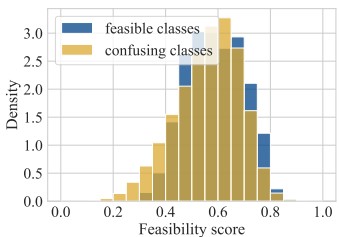

Figure 6: Feasibility scores for question-answer "yes" format in the in-context prompt.

Similar trends are observed for "Format QA: score" where we use a question-answer format with guiding LLMs to respond with an integer score instead of a binary answer. To obtain an integer score as an answer from the LLMs, we construct the guidance component of the human message as:

```
"The following list consists of words and their likelihood of existence
 in the real world, scored on a scale of 0 to 9.
 − {s₁} {o₁}, score:  9
 − {s₂} {o₂}, score:  9
 − ...
 − {sₙ} {oₙ}, score:  9
 What is the score for "{s} {o}"?"
```

If we had access to a more nuanced classification of prior feasibility scores for the seen classes, we could provide the LLM with a more informative guidance. As this is not available, we had to choose fixed score value. The results are shown in the "Format QA: score" row of Table 3 in the main text. We observe the performance drops across the datasets, e.g. 17.4%→12.3% and 6.55%→3.69% in harmonic mean on MIT-States and C-GQA. These observations, together with "QA: yes" format, indicate that employing a list format for guidance rather than a question-answer format is crucial in obtaining accurate feasibility scores.

---

[2] https://github.com/BatsResearch/csp

## E    QUALITATIVE EXAMPLES.

| MIT-States | Method | | | C-GQA | Method | | |
|---|---|---|---|---|---|---|---|
| class | GloVe | ConceptNet | FLM (ours) | class | GloVe | ConceptNet | FLM (ours) |
| folded book | ✘ | ✔ | ✔ | blue table | ✔ | ✘ | ✔ |
| rusty truck | ✘ | ✔ | ✔ | brown cake | ✔ | ✔ | ✘ |
| small dog | ✘ | ✘ | ✔ | balding person | ✘ | ✔ | ✔ |
| eroded granite | ✔ | ✔ | ✘ | blue tray | ✘ | ✘ | ✔ |
| gray stove | ✔ | ✘ | ✔ | asian person | ✘ | ✔ | ✔ |
| thick ring | ✘ | ✘ | ✘ | yellow leaf | ✘ | ✔ | ✔ |

Table 4: Qualitative examples from the MIT-States and C-GQA datasets. A state-object pair is deemed feasible (✔) or infeasible (✘) by the respective methods.

We examine qualitative examples of feasibility classifications where we compare the predictions of three methods in Table 4. It displays unseen classes most commonly found in the test datasets. We observe that in both the MIT-States and C-GQA datasets, our FLM and ConceptNet tend to predict the given classes as feasible more frequently compared to GloVe. One interesting observation arises from the class "small dog" which is surprisingly predicted as infeasible in both GloVe and ConceptNet, but correctly identified as feasible by FLM.

## F    PSEUDO-CODE FOR FLM.

We present the pseudo-code for the implementation of our proposed Feasibility with Language Model in Listing 1.

```python
# prompt generator
class PromptToLLM():
    def __init__(self, model_path, train_pairs, max_n_pos):
        self.model_path = model_path
        self.train_pairs = train_pairs
        self.max_n_pos = max_n_pos

        self.persona = "You are a helpful, respectful and honest assistant."
        self.instruction = "Answer with a single word, yes or no."
        self.sysmsg = self.persona + " " + self.instruction
        self.pre_guidance = "The following list consists of word combinations that make sense."
        self.positive_guidance = "\n- {0} {1}"
        self.query_template = "Does \"{0} {1}\" fit into the list above?" # 0: state, 1: object

    def get_in_context_list(self, pair):
        q_attr, q_obj = pair
        pos_params = \
            [(_a, _o, indef(_a), indef(_o))
            for _a, _o in self.train_pairs if _a == q_attr or _o == q_obj]
        if len(pos_params) > self.max_n_pos:
            pos_params = random.sample(pos_params, k=self.max_n_pos)

        in_context = \
            self.pre_guidance + \
            " ".join([self.positive_guidance.format(*param) for param in pos_params])
        in_context = in_context.strip()
        if len(in_context) != 0:
            in_context += "\n"
        return in_context

    def get_human_messages(self, pairs):
        human_messages = []
        for pair in pairs:
            query = self.query_template.format(*pairs)
            in_context = self.get_in_context_list(pair)
            human_message = " ".join([in_context, query])
            human_messages.append(human_message.strip())
        return human_messages

    def get_conv_prompts(self, human_messages):
        prompts = []
        for hmsg in human_messages:
            conv = get_default_conv_template(self.model_path).copy()
            conv.set_system_message(self.sysmsg)
            conv.append_message(conv.roles[0], hmsg)
            conv.append_message(conv.roles[1], None)
            prompt = conv.get_prompt()
```

```
48              prompts.append(prompt)
49          return prompts
50
51      def __call__(self, pairs):
52          human_messages = self.get_human_messages(pairs)
53          prompts = self.get_conv_prompts(human_messages)
54          return prompts
55
56  # model inference
57  def get_yes_no_logits(model, tokenizer, prompts):
58      tokenized = tokenizer(prompts, padding="longest",
59          return_attention_mask=True, return_tensors='pt').to(model.device)
60      out = model(**tokenized)
61      logits = out.logits
62
63      # Indices of "Yes" and "No" in Vicuna's token embedding: 3869 and 1939
64      last_token_logits = logits[:, -1, :]
65      return last_token_logits[:, [1939, 3869]]
66
67  if __name__ == "__main__":
68      model_path = LLM_MODEL_PATH
69      train_pairs = SEEN_PAIRS_IN_TRAINING_DATASET
70      ow_pairs = OPEN_WORLD_PAIRS
71      batch_size = BATCH_SIZE
72      max_n_pos = MAXIMUM_NUMBER_OF_POSITIVES_IN_IN_CONTEXT_PROMPT
73
74      # Load LLM
75      model, tokenizer = load_model(model_path)
76      tokenizer.pad_token = tokenizer.eos_token
77      tokenizer.padding_side = "left"
78
79      # prompt instance
80      get_prompts = PromptToLLM(model_path, train_paris, max_n_pos)
81
82      # Feasibility with Language Models (FLM)
83      scores = []
84      for batch_idx in np.arange(0, len(ow_pairs), batch_size):
85          query_pairs = ow_pairs[batch_idx : batch_idx + batch_size]
86          prompts = get_prompts(query_pairs)
87          _scores = get_yes_no_logits(model, tokenizer, prompts)
88          scores.append(_scores)
```

Listing 1: Pseudo-code for FLM.

