# OpenReview forum: "Feasibility with Language Models for Open-World Compositional Zero-Shot Learning"
_ICLR.cc/2024/Conference — Submitted to ICLR 2024_

### Official Review · Reviewer_Z7Er · 2023-10-30

**Soundness:** 3 good
**Presentation:** 3 good
**Contribution:** 3 good
**Rating:** 5
**Confidence:** 5

**Summary:**

This paper proposes to use Large Scale Language Model to identify infeasible pairs for open world compositional zero shot learning (ow-czsl). Speicifically, the authors propose In-context Learning to ensure the feasibility of the composition, which utilize a few examples of true feasible pairs, allowing the LLMs to learn from these instances and better understand what constitutes feasibility within the dataset. The experiments demonstrate that the proposed method improves over the existing methods that identify compositions.

**Strengths:**

1.The paper is well-written and easy to follow.
2.According to the experiments, FLM achieves noteworthy improvement in performance.

**Weaknesses:**

1.Time costs need to be taken into account. As is known to all, the open-word setting will produce a large number of virtual compositions, which will bring a huge amount of calculation to the model (the proposed model process all possible pairs once and predict the score).
2.According to the paper, the In-context Learning seems not to be trained in the process, which means that the performance relies on the LLMs. However, there existing some objects and states that are totally unknown to LLMs. In this situation, the proposed model cannot transfer the knowledge to the unknown compositions.
3.The proposed method relies much on the quality of LLMs, and the transferability of the model is not reflected in the paper.

**Questions:**

NA

---

> ### Author Response · Authors · 2023-11-16
> **Author Rebuttal by Authors**
>
> We would like to thank the reviewer for their insightful comments and questions as well as for pointing out that FLM achieves noteworthy improvement in CZSL performance. We address the reviewer’s questions below.
>
> > Time costs need to be taken into account. As is known to all, the open-word setting will produce a large number of virtual compositions, which will bring a huge amount of calculation to the model (the proposed model process all possible pairs once and predict the score).
>
> For the MIT-States dataset, obtaining feasibility scores for 28,175 pairs with Vicuna 13B takes approximately 40 minutes on a single A100 GPU with a batch size of 16. For the UT-Zappos dataset, obtaining scores for 192 pairs takes about 1 minute with a batch size of 16. Finally for the CGQA dataset, obtaining scores for 278,362 pairs takes approximately 250 minutes with a batch size of 5 (as it requires longer input sequences due to more in-context examples). Note that this is a one-time preprocessing step and does not affect inference time. Note that our method does not require any training, which would otherwise incur a time cost and it allows to update a previously computed list of state-objects pairs with additional states/objects without invalidating previously computed scores (because no re-training is required).
>
> > According to the paper, the In-context Learning seems not to be trained in the process, which means that the performance relies on the LLMs. However, there existing some objects and states that are totally unknown to LLMs. In this situation, the proposed model cannot transfer the knowledge to the unknown compositions.
>
> LLMs have a vast vocabulary that includes all practically relevant states and objects considered in CZSL datasets (fine-grained shoe dataset UT-Zappos, common real-world objects in MIT states and C-GQA) and their language understanding is generally more comprehensive compared to GloVe and ConceptNet. Nonetheless, when we test the LLM with unknown words (e.g. made-up), it defaults to considering their pairs as infeasible unless their feasibility is very clear from the in-context example, i.e., it cannot as easily relate objects or states by their semantic similarity.
>
> When new states and objects become available (e.g. neologism or recent product names), any model used for feasibility prediction (LLMs for FLM, GloVe, ConceptNet) would have to be updated with relevant training data. In the case of FLM, we would likely use the most recently available LLM. For anything in-distribution (e.g. real-world states/objects), we find that in-context examples are effective in adjusting the LLM to the specific task without requiring costly re-training.
>
> > The proposed method relies much on the quality of LLMs, and the transferability of the model is not reflected in the paper.
>
> As explained in our previous answers, we believe that in-context learning allows LLMs to transfer and adapt to new datasets, which has also been shown by Brown et al [A]. Our ablations in Table 3 show how the performance of directly applying the model without in-context learning is much worse (e.g. dropping from 17.4% to 15.4% in harmonic mean on MIT-States). We see this as a strength of our approach as the LLM can be readily replaced by a more capable model in the future without requiring task-specific fine-tuning.
>
> [A] Language models are few-shot learners. NeurIPS, 2020.

---

### Official Review · Reviewer_niw2 · 2023-10-31

**Soundness:** 2 fair
**Presentation:** 3 good
**Contribution:** 2 fair
**Rating:** 6
**Confidence:** 4

**Summary:**

This paper proposes a novel feasibility calibration scheme for open-world compositional zero-shot learning. The key ideas leverage large language models as the intermediate agency for feasibility decisions. The authors conducted extensive experiments to show that the proposed scheme can improve compositional zero-shot recognition. The authors also include ablation experiments to address effects of underlying LLMs and prompts.

**Strengths:**

To the best of the reviewer’s knowledge, this method proposed in this paper is novel. This paper is clearly motivated and the intuition behind the proposed methods are also very clear. The idea of using LLMs for solving feasibility conflicts is simple yet quite effective. The authors also show that as an orthogonal component to existing compositional zero shot learning methods, LLM-guided feasibility calibration can clearly boost the performance for most of the scenarios.

**Weaknesses:**

Despite the work’s obvious merit, the idea itself is very simple. Within the ablations, it would be helpful if the authors are to thoroughly examine more variants of prompts since LLMs output can vary a lot. The performance variations under such scenarios would be very informative to the community.

**Questions:**

Despite the method being effective as the authors have shown, the idea of using a single LLM for decisions may suffer from the common issues of hallucinations. I wonder if the authors have tapped into potential solutions to get around this issue.

---

> ### Author Response · Authors · 2023-11-16
> **Author Rebuttal by Authors**
>
> We thank the reviewer for their helpful comments and suggestions. We appreciate that the reviewer highlights the novelty of FLM and that our conceptually simple method is effective in boosting existing CZSL methods.
>
> > Despite the work’s obvious merit, the idea itself is very simple.
>
> We agree with the simplicity of our FLM method. However, it is important to emphasize that simplicity doesn’t negate the effectiveness where our method consistently improves the performance across different benchmarks and various CZSL models. Moreover, its versatility enables the integration with any existing CZSL models.
>
> > Within the ablations, it would be helpful if the authors are to thoroughly examine more variants of prompts since LLMs output can vary a lot. The performance variations under such scenarios would be very informative to the community.
>
> To examine a broader range of prompt variations, we added two more instruction sentences, i.e. “Answer with yes or no.” and “Answer with yes or no, followed by an explanation”, and one more query sentence, i.e. “Considering the list above, does “{s} {o} align with the contents?”. By adding these sentences to the list described in Appendix B, we experiment  with 64 (=4^3) different prompts. We report the results on MIT-States as a box plot in Figure 5 in Appendix B. We first observe the results vary with different prompts. For instance, the Harmonic mean accuracy in CSP ranges from 15.5% to 18.1%. However, despite this variability, most of the prompts outperform the baselines. For instance, the Glove result always lies close to or below the lower quartile (25th percentile) of the box plot, and ConceptNet even further below that.
>
> > Despite the method being effective as the authors have shown, the idea of using a single LLM for decisions may suffer from the common issues of hallucinations. I wonder if the authors have tapped into potential solutions to get around this issue.
>
> As we only probe the LLM for the tokens “Yes” and “No”, FLM is not directly affected by LLMs making up facts otherwise commonly observed as hallucinations. In the context of feasibility prediction, one could most closely relate this to making an incorrect prediction, e.g., assuming a pair to be infeasible although it exists in the dataset. From our ablation study in Table 3, we find that the in-context examples contribute the most to mitigating prediction errors, but it is not straightforward to identify mistakes as “hallucinations” or any other limitation of the LLM.

---

### Official Review · Reviewer_655z · 2023-11-01

**Soundness:** 3 good
**Presentation:** 2 fair
**Contribution:** 1 poor
**Rating:** 3
**Confidence:** 4

**Summary:**

This paper aims to determine the feasibility of state-object (s, o) combinations (i.e., while the phrase“hot fire” is feasible, the phrase“wet fire” is not feasible). This paper proposed to use large language models (LLM), such as Vicuna and ChatGPT to classify whether the phrase is feasible or not. This is implemented by entering ``Does a/an [THE PHRASE] exist in the real world?” as the input to the LLM and the probability of LLM answering “yes” is the feasibility score. The author also tries to show several feasible phrases to the LLM to help the LLM decide whether [THE PHRASE] is feasible or not.

**Strengths:**

1. Figure 1 is well designed and helps the reader to understand the content.
2. The proposed method is simple and easy to understand.

**Weaknesses:**

1. The main concern of this work is its contribution. The paper basically uses the existing LLM to determine the feasibility of a state-object combination. This only shows that the existing LLM is able to determine the feasibility of a state-object combination, but what is the author’s contribution throughout the process?
2. Since different threshold will affect the binary classification performance, wouldn’t a metric like ROC curve suits the tasks better?
3. For Figure 2, it seems that both green and red block only show the feasible (s,o) pairs. The author is suggested to show some infeasible (s,o) pairs and the model prediction on those infeasible (s,o) pairs.
4. It is challenging to tell whether GloVe or the proposed FLM separates the feasible and infeasible better by only looking at the figures. The author is suggested to show some numerical results to support the claim.
5. In the evaluation metric, the author mentioned that the calibration bias is varied. Does it mean that different calibration bias is used for different metric?
6. Typo: such “at” ChatGPT

**Questions:**

1. For the rebuttal, the author is suggested to highlight the contribution of this work. After reading the submission, this paper is more like showing the observation that existing LLM already did a great job on identifying infeasible (s,o) pairs. There is no modification of the LLM, and no loss function is proposed.
2. Some clarification problem in the weakness section needs to be addressed.

---

> ### Author Response · Authors · 2023-11-16
> **Author Rebuttal by Authors**
>
> We thank the reviewer for their thorough evaluation and for acknowledging that our method is simple and easy to understand. We address the reviewer's questions below.
>
> > The main concern of this work is its contribution. The paper basically uses the existing LLM to determine the feasibility of a state-object combination. This only shows that the existing LLM is able to determine the feasibility of a state-object combination, but what is the author’s contribution throughout the process?
>
> To the best of our knowledge, we are the first to leverage LLMs and in particular their in-context learning capability to enhance the CZSL performance. Though simple, our FLM substantially improves CZSL performance. Moreover, our FLM is an orthogonal work that can be integrated with any CZSL methods to boost performance in the open-world setting. By replacing an important step in the CZSL pipeline, our approach has an impact on both present and future CSZL methods. This approach also has been acknowledged as innovative by reviewer niw2.
>
> > Since different threshold will affect the binary classification performance, wouldn’t a metric like ROC curve suits the tasks better?
>
> We report AUC of the CZSL performance in Table 1 which represents the ROC curve as we vary the calibration bias. In Table 2, we then report the feasibility accuracies for the fixed threshold that is used in Table 1 to make a direct connection to the downstream task of CZSL.
>
> > For Figure 2, it seems that both green and red block only show the feasible (s,o) pairs. The author is suggested to show some infeasible (s,o) pairs and the model prediction on those infeasible (s,o) pairs.
>
> In the CSP model on MIT-States dataset, our FLM scores “peeled sauce” and “molten milk”, which are infeasible pairs, as -0.46 and –0.30, respectively, while GloVe's scores are 0.1 and 0.18 and ConceptNet's scores are 0.09 and 0.05. However, there are also “failure cases” in predicting an infeasible pair. For example, “mashed tomato” (0.04, 0.28, 0.30 by FLM, GloVe, and Conceptnet, respectively) and “fresh apple” (0.04, 0.02, 0.07), though not included in the closed-world ground-truth pairs, are considered feasible by all methods. One could argue that those are correct predictions, but in the context of the dataset they become confusing pairs for the model as there are no related images in the dataset. It is difficult to find cases where a human would clearly consider a pair to be infeasible, but FLM predicts it to be feasible. Examples that come closest are “cracked shore” or “dented building”. As mentioned in Section 4.3, it is important to note that some realistically feasible pairs are simply not included in the dataset and, thus, we cannot show images for any infeasible pairs from the datasets.
>
> > It is challenging to tell whether GloVe or the proposed FLM separates the feasible and infeasible better by only looking at the figures. The author is suggested to show some numerical results to support the claim.
>
> As shown by the numerical results in Table 2 that support Figures 2 and 3, our FLM performs the best on either feasible or infeasible accuracies. Considering both metrics together through arithmetic and harmonic means, our method best separates the feasible and infeasible pairs across all datasets.
>
> > In the evaluation metric, the author mentioned that the calibration bias is varied. Does it mean that different calibration bias is used for different metric?
>
> Yes, we follow Nayak et al. (CSP) in that the calibration bias is different for every metric. The calibration bias, which is subtracted from the CZSL model outputs of the seen classes but not the unseen classes, is varied to plot the ROC curve of the seen class and unseen class accuracies. The seen class accuracy (metric S) corresponds to the minimal bias value, the unseen class accuracy (metric U) aligns with the maximum bias value, and the H metric is when the bias value reaches the best harmonic mean between the seen class and unseen class accuracies. The AUC metric, measuring the area under the ROC curve, remains unaffected by the value of the calibration bias as it takes all possible values into account.
>
> > Typo: such “at” ChatGPT
>
> Thank you for pointing this out. We rectified the typo.

---

> ### Comment · Reviewer_655z · 2023-12-04
> **Thanks for the rebuttal**
>
> Thanks to the authors for providing the rebuttal. I carefully read all the reviews and rebuttals. I echo with Reviewer iHMU that the contribution of this paper is marginal, and thus I am prone to keep my rating to this paper.

---

### Official Review · Reviewer_iHMU · 2023-11-02

**Soundness:** 2 fair
**Presentation:** 2 fair
**Contribution:** 2 fair
**Rating:** 3
**Confidence:** 4

**Summary:**

In this paper, the authors propose a novel approach that leverages large language models (LLMs) to predict the feasibility of the state-object pair for the open-world compositional zero-shot learning (OW-CZSL). A prompt is designed to query the feasibility score by leveraging the autoregressive nature of LLMs.

**Strengths:**

S1: The studied problem about open-world compositional zero-shot learning is significant important and can apply to the real-world scene.

S2: The large-language models are used to reduce the gap between machines and humans.

S3: Extensive experiments on many prompt variants and six LLMs shows the best performence.

**Weaknesses:**

W1: Is this the first paper to solve the CZSL problem by using  the LLMs? If yes, I am curious about the motivation or some motivation experiments to demonstrate the effectiveness of LLMs? If no, I tend to see some differents compared with other published related works.

W2: This method in this paper is not novel and performance improvement depends entirely on the language model. If the language model introduces biases, such as racial discrimination, during training, will this also affect downstream tasks?

W3: Does a more powerful language model perform best in this paper?

In my opinion, simply introducing a language model to solve downstream tasks does not reach the upper limit of ICLR acceptance.

**Questions:**

See Weaknesses

---

> ### Author Response · Authors · 2023-11-16
> **Author Rebuttal by Authors**
>
> We thank the reviewer for their insightful comments that help us improve the paper and for pointing out that we study an important problem where we show with extensive experiments that our LLM-based method achieves the best performance.
>
> > Is this the first paper to solve the CZSL problem by using the LLMs? If yes, I am curious about the motivation or some motivation experiments to demonstrate the effectiveness of LLMs?
>
> To the best of our knowledge, we are the first to leverage LLMs and in particular their in-context learning (ICL) ability to enhance CZSL. Contextual understanding of LLMs would enhance their ability to predict feasibility, as illustrated by the example of “dark fire” in the Introduction section. This motivated us to use the ICL capability of LLMs rather than a simple prompt, resulting in a substantial improvement in CZSL performance. By replacing an important step in the CZSL pipeline, our approach has an impact on both present and future CSZL methods. This approach also has been acknowledged as innovative by reviewer niw2.
>
> > I tend to see some differents compared with other published related works.
>
> We tackle the issue of feasibility prediction while most other published works focus on changing the model pipeline (model architecture or loss function) to improve the performance. We argue that our FLM is an orthogonal work that can be integrated with any CZSL methods to boost performance in the open-world setting.
>
> > This method in this paper is not novel and performance improvement depends entirely on the language model. If the language model introduces biases, such as racial discrimination, during training, will this also affect downstream tasks?
>
> This is a valid concern. If biased attributes are involved in the composition, the LLMs’ response for feasibility prediction may be affected, leading to a wrong prediction. Understanding and addressing biases in the LLMs could be important for the CZSL downstream task. The general benchmarks in CZSL task, i.e. MIT-States, UT-Zappos, and C-GQA, involve common objects and states (e.g. shoes or animals, colors or conditions) without including any humans. These state-object pairs are not typically considered social biases, such as racial discrimination. Hence, these states might not be biased in the same way as human-related attributes, such that biases in LLMs could have a smaller effect on CZSL downstream tasks. Nonetheless, investigating the effect of biases in LLMs to the CZSL downstream task would be an intriguing open question.
>
> > Does a more powerful language model perform best in this paper?
>
> The general trend of CZSL downstream performance appears to align with the language model’s capabilities. In Figure 4, particularly on MIT-States, we observe that ChatGPT and Claude-2 outperform Vicuna-13B with LLaMA-2 being the last in the binary setup, reflecting a ranking similar to the LLM leaderboard. However, we also observed the exception where GPT-4 did not follow the expected trend in the CZSL downstream task.

---

> > ### Comment · Reviewer_iHMU · 2023-11-22
> > **Thank you for authors‘ the detailed response**
> >
> > Thank you for authors’ the detailed response.
> >
> > I have carefully read the author's response and the comments of other reviewers. I still feel that the main contribution of this paper in introducing the LLM is insufficient. Therefore, I keep my score.

---

### Meta-Review · Area_Chair_bAkY · 2023-12-17

**Metareview:**

The authors introduce an LLM based approach to reason over object states -- effectively querying for commonsense knowledge about the world.  The approach leverages ICL to assess several models and demonstrate that their approach to querying state-object information and weighting feasibility queries is more effective than lexical embeddings or KBs (GloVE and ConceptNet)

**Justification For Why Not Higher Score:**

Reviewers simply did not feel they learned enough from this work and that the results were predictable.  A more comprehensive set of experiments and implications for the field would strengthen the work.

**Justification For Why Not Lower Score:**

N/A

---

### Decision · Program_Chairs · 2024-01-16

Reject